# Conceptualizing Family Influences on Children’s Energy Balance-Related Behaviors: Levels of Interacting Family Environmental Subsystems (The LIFES Framework)

**DOI:** 10.3390/ijerph15122714

**Published:** 2018-12-01

**Authors:** Christina Y.N. Niermann, Sanne M.P.L. Gerards, Stef P.J. Kremers

**Affiliations:** 1Department of Sport Science, University of Konstanz, P.O. Box 30, 78457 Konstanz, Germany; 2Department of Health Promotion, NUTRIM School of Nutrition and Translational Research in Metabolism, Maastricht University, 6200 MD Maastricht, The Netherlands; sanne.gerards@maastrichtuniversity.nl (S.M.P.L.G.); s.kremers@maastrichtuniversity.nl (S.P.J.K.)

**Keywords:** health behavior, physical activity, eating behavior, sedentary behavior, family environment, family systems, parents, parental behavior

## Abstract

Healthy or unhealthy behavioral patterns develop and are maintained in a family context. The importance of the family environment for children’s and adolescents’ energy balance-related behaviors (EBRBs) has been shown previously. However, the way different family environmental factors are interrelated and interact with personal factors (e.g., motivation) are not well understood. Furthermore, the majority of studies have focused on the parent-child subsystem. However, there are family-level socialization dynamics that affect the development of a healthy lifestyle beyond the impact of parenting behaviors. The current paper aims to synthesize theoretical and empirical literature on different types of family influences. The Levels of Interacting Family Environmental Subsystems (LIFES) framework incorporates family influences on three levels (immediate, proximal, distal) and of three subsystems (individual, parent-child, family), relates them to each other and postulates potential paths of influence on children’s EBRBs. Several studies examining specific sections of the framework provide empirical support for LIFES’ propositions. Future studies should place their research in the context of the interrelationship of different family environmental influences. A better understanding of the interrelated influences would enhance the understanding of the development and maintenance of overweight and obesity among children and is crucial for the development of effective interventions.

## 1. Background

Too many children and adolescents make unhealthy food choices, are not sufficiently physically active and spend too much time in sedentary behaviors [1,2,3]. Research has shown that these behaviors, referred to as energy balance-related behaviors (EBRBs) [4], are associated with negative effects on health and well-being [5,6,7,8,9,10], as they have been associated with weight gain and the development of overweight and obesity. Since unhealthy lifestyle habits often track into adulthood [11], it is important that healthy lifestyle habits are established at a young age.

It is well-known that individuals’ healthy lifestyles are determined by a multitude of influences, including individual as well as social and physical environment factors and their interplay. In this paper, we address the relevance of the family microsystem. The family is the basic social context where healthy or unhealthy behavior patterns develop and are maintained [12]. It serves as a reference point for the development of behavioral habits. Family life implies a large number of health-related cues, as daily family routines, such as family meals, choice and preparation of food and communication, are an inherent part of family life [12,13]. Furthermore, specific formal or informal rules may develop, which stimulate individuals’ eating or activity and inactivity patterns [12]. The family is an instance of control and organization which provides socio-emotional support [14], and internalized concepts of health and well-being, values, attitudes and self-perception of competences are formed within the family [15,16]. The majority of studies in the field of child EBRBs have focused on the influence of parents on their children. However, the family is more than interactions between parent and child. Family-level socialization dynamics (arising for example from family functioning, cohesion within the family, communication patterns between family members, shared values and attitudes) may affect the development of a healthy lifestyle beyond the impact of parenting behaviors. The current paper aims to synthesize theoretical and empirical literature on different types and levels of family influences and to describe their assumed interrelationships by developing a framework called Levels of Interacting Family Environmental Subsystems (LIFES). LIFES aims to organize the variety of different family environmental factors and environment—behavior pathways in youth. It helps to advance the research field as it supports researchers to derive specific hypotheses regarding underlying mechanisms of family environmental influences on individual’s behaviors. LIFES-based research might inform future interventions as it reveals processes that impede or support individual behavior change. It could also help to find answers for example regarding the question how the family environment should be integrated in intervention programs in order to increase sustained effects on children’s and adolescents’ EBRBs.

### Family and EBRBs of Children

There is a large body of research showing the relevance of the family environment for children’s and adolescents’ EBRBs. Research into different forms of parental influences on children’s and adolescents’ behavior has shown that parents play an important role in the development of a healthy lifestyle [17,18]. Parents are “gate keepers” of healthful eating and engaging in physical activities [19,20]. Mechanisms that have been studied include parenting practices such as modeling, monitoring, and encouragement [21,22,23,24,25,26] and more general concepts such as parenting styles or general parenting [27,28,29,30]. Overall, these studies reflect the important role of the parents in the development of a healthy lifestyle by children and adolescents.

The traditional approach to investigating family influences on children’s health or health-related behavior is unidirectional. Studies on parental influences often assign a passive role to children, seeing children mostly as recipients of parenting behaviors. However, recent studies also provide evidence that parenting behaviors occur in response to the child-level factors and children’s behavior. These studies showed that parent’s choices regarding food parenting are influenced by the child’s weight status and behaviors [31,32,33,34,35]. Despite such findings, the literature is still dominated by studies characterizing parents as the agents acting upon the child, without taking reciprocal processes or feedback loops into account.

The majority of studies in this field have focused on the parent-child subsystem [20,36,37]. However, in this paper we will argue that the family is more than the interactions between parent and child. We assume that family-level socialization dynamics [38] affect the development and maintenance of a healthy lifestyle [37], beyond the impact of parenting behaviors. For example, interpersonal behaviors that occur at the family level, such as family meals, might have an influence on EBRBs [36]. Research has consistently shown that having frequent family meals is associated with a number of health benefits for children e.g., lower BMI and healthy dietary intake [36]. Another example is family functioning, which comprises structural and organizational properties of the family and interpersonal interactions. Family functioning is reflected in aspects like communication patterns, role fulfillment, adaptability, management of conflicts, involvement, warmth/closeness, and behavior control. This topic has been addressed by a small number of studies, which found that family functioning may be a relevant correlate of EBRBs [36,39,40].

Gaining knowledge about the interplay of family environmental factors is crucial to understand the development and maintenance of children’s and adolescents’ ERBRs [37]. It should be noted that although the family environment is important for both children and adolescents, the relevance of specific family environmental factors will depend on the age of the child. Therefore, these age groups should be studied separately, or, at least age should be incorporated not only as a confounder but also as a potential moderator. In the current paper, we seek to bring together findings reported in theoretical as well as empirical literature on different types and levels of family influences, and we describe their assumed interrelationships. Since there is currently no framework that addresses such a “bigger picture” of family environmental influences, we have developed a framework, called Levels of Interacting Family Environmental Systems (LIFES) with the aim of (1) illustrating that the family is more than parent-child interactions, (2) assisting researchers in putting their study in the context of the family environment, (3) stimulating research that includes higher level (upstream) influences, and (4) identifying areas of research that are in need of empirical research.

## 2. Levels of Interacting Family Environmental Systems (LIFES)

### 2.1. Theoretical Background

The development of the LIFES framework was based on several theoretical approaches, such as ecological approaches [41,42], systems theories [43], and Family Systems Theory [44,45]. Ecological approaches to health behavior propose that individuals are embedded in physical and sociocultural surroundings. Factors are typically defined on different levels of influence on individual health-related behaviors (e.g., intrapersonal, interpersonal, organizational, community, and policy) and these factors are hypothesized to interact. LIFES focus on the family as an interpersonal level and the interactions with intrapersonal factors. The family can be considered a complex adaptive system [46]. In this view, an individual functions in a hierarchical system of elements, from micro-level to macro-level. An essential part of this paradigm is that the operating components at all levels of the person-environment system function and develop as an integrated system. In actual operation, the role and functioning of each element depend on its context of other, simultaneously operating components, both horizontally (i.e., within levels) and vertically (i.e., across levels). In the operationalization of the family context, we thus adopted principles from Dynamic Systems Theories [47]. The concept of system refers to a “complex of interacting elements” [43] or a “group of parts that are interacting according to some kind of process” [48]. What is common to the various definitions of a system is not the characteristics of the individual units or parts but rather the extent and nature of linkages or interrelationships among the various units [49]. The impact of a system is more than just the sum of the individual parts. The functioning of any one element in a system depends on the existence and operation of other elements in the system. This implies that the impact of the fruit bowl on the kitchen table cannot be understood by mechanistically modeling it, correcting for all other potential determinants in the causal chain (e.g., accessibility of the fruit, personal attributes such as impulsivity and gender), but requires examining the system conditions under which the fruit bowl has an impact [50]. Accordingly, Family Systems Theory describes the family as a complex interacting system and an organized whole. Family Systems Theory assumes that (1) the elements of the system are interconnected and interdependent, (2) the system is best understood when viewed as a whole, (3) subsystems exist within the family system (e.g., individuals, parent-child subsystem), (4) the behavior of the system interacts in a feedback loop with the environment, (5) the systems are not reality, but heuristic models for understanding, (6) systems strives to maintain equilibrium and (7) should be able to adapt and evolve with changes [44,45].

By incorporating core assumptions of ecological and systems theories, the LIFES framework aims to conceptualize the content and interactive nature of different types and levels of family environmental influences on children’s and adolescents’ EBRBs. It covers three subsystems within a family and three levels of environmental factors ranging from behavior-specific to general, and the framework proposes specific postulations on the interactions between these 3 × 3 sectors (Figure 1).

### 2.2. General Description

The framework proposes that children’s and adolescents’ EBTBs are affected by child factors (e.g., motivational, volitional and cognitive factors, personality, gender) (highlighted blue in Figure 1) and by factors arising from the family (e.g., parents’ behaviors, parenting practices, family functioning) (marked yellow in Figure 1). In order to reduce the complexity, the framework incorporates the individual (child, parent) and parent-child subsystem and the family system. However, according to Family Systems Theory further subsystems exist, that were not focused in this framework (e.g., siblings, spouse subsystem). Furthermore, LIFES differentiates three levels of children’s and family environmental influences (immediate, proximal and distal).

Therefore, LIFES comprises:three (sub)systemsindividual: influences related to individual family members (e.g., child, mother, father)parent-child: influences related to parent-child interactions (e.g., mother-child, father-child)family: influences related to the family as a wholeand three levels of influencesimmediate: manifested behaviorsproximal: behavior-specific factorsdistal: general factors

The ultimate outcomes in this framework are children’s and adolescents’ EBRBs. On the one hand children’s EBTBs are influenced by factors on the proximal (e.g., self-efficacy regarding being physically active) and distal level (e.g., general self-efficacy) within the child-subsystem. On the other hand, the child is one subsystem within the family system. Therefore, understanding of child’s behavior and influencing factors requires taking into account other subsystems and the family system. In order to understand child’s EBRBs the framework focus on the interactions between the child subsystem, the parent subsystem, the parent-child subsystem, and the family system.

It should be noted that family is an inhomogeneous social context, there is no narrow definition in this research field. A family could be described as a complex interacting system and an organized whole. This description includes natural parents as well as other caregiving members such as step-parents or grandparents as part of the family system. Families also incorporate siblings and there is growing evidence regarding the important role of sibling relationships (e.g., [51,52,53]).

It should be noted that the framework illustrated in Figure 1 is a vast simplification as it is not possible to illustrate the complexity of the influences within the family system. Figure 1 illustrates that child’s EBRBs and factors on the proximal and distal level are interrelated to factors of the other (sub)systems: parent (mother and/or father), parent-child (mother-child and/or father-child), and family. Within all (sub)systems there are factors on equivalent levels—immediate, proximal, and distal. Therefore, factors arising from the level-subsystem combinations (e.g., proximal factors within the parent-child subsystem) are interrelated with the equivalent child levels (i.e., proximal influences on EBRBs within the child-subsystem). The level-subsystem combinations are not limited to one single factor, but several factors could be allocated to each (e.g., distal influences of the parents can comprise factors such as personality, educational level, general self-efficacy; proximal influences of parent-child can comprise: perceived importance of child’s physical activity, perceived competence of the child regarding physical activity, cognitions regarding the provision of healthy foods).

We assume reciprocal influences between the subsystems and between the levels. Depending on the position in the model, the influences are supposed to be either direct or indirect, through mediators or moderators. Direct links are supposed to exist between “neighbors” within subsystems or within levels. The other links are supposed to be mediated. In addition, higher-level moderation effects are assumed.

Below we describe the sectors of the LIFES framework (see Figure 1), representing the combinations of subsystems (individual—a, parent-child—b, family—c) and levels (immediate—1, proximal—2, distal—3) of the LIFES framework. In addition, relevant empirical examples are presented for each sector.

### 2.3. Description of Levels and Subsystems

#### 2.3.1. Immediate Level

The factors at the immediate level are assumed to be directly linked to children’s and adolescents’ EBRBs.

##### Individual—Parents’ Health Behavior (1a)

Individual behaviors of family members are reciprocally related within families [54,55]. Parents’ health behaviors (health behaviors of mothers and/or fathers) are assumed to exert a passive influence on children’s and adolescents’ EBRBs [56]. However, some of these behaviors take place in front of the children, and could therefore be labeled as modeling. It should be noted that there is a conceptual difference between modeling and parental role modeling. Parental role modeling is a parenting practice and therefore an active and intentional process, whereas modeling is parents’ behavior which unintentionally takes place in front of the children [25,56].


*Association with children’s EBRBs*


Several studies found that parents’ dietary intake was found to be positively related to children’s fruit and vegetable intake [17,57,58], fat intake [17], breakfast consumption [59], and (although less consistently) soft drink consumption [17,28,59].

Similarly, parents’ physical activity level seems to be a significant correlate of children’s physical activities [17,59,60,61,62]. Additionally, sedentary time, such as TV viewing, was found to be associated with adolescents’ sedentary time [59,63]. However, regarding dietary intake Wang and colleagues [64] concluded in their systematic review and meta-analysis that overall the associations are weak. They assumed that the effect of parents’ dietary intake should not be examined isolated from other variables. They concluded that the dietary intake of children is affected by a complex interaction of variables at different levels.

##### Parent-Child—Parenting Practices (1b)

Parenting practices are goal–directed parental behaviors with the aim of influencing children’s behaviors [65], including children’s nutrition and physical activity behaviors. Parenting practices are behavioral strategies related to how much, when and what children eat and to what extent they are physically active. A broad range of parenting practices have been investigated; we refer to Gevers and colleagues [26,66], O’Connor and colleagues [67], Mâsse and colleagues [68], Davison and colleagues [69] and Vaughn and colleagues [70] for elaborate overviews and measures of different parenting practices. An example of a parenting practice regarding physical activity is providing logistic support to the child’s activity, e.g., taking the children to places where they can be physically active [68,71]. An example of a food-related parenting practice is the parental role modeling of healthy food intake [67].


*Association with Children’s EBRBs*


A large number of studies have investigated the association between parenting practices and children’s food intake, which have been summarized in various systematic reviews (for example [59,72]. Some parenting practices were found to be consistently associated with children’s nutrition intake, while the effect of other parenting practices was less consistent. To illustrate this, availability and accessibility at home has consistently been associated with positive effects on children’s healthy food intake [17,57,58]. The effects of restriction, pressure to eat and permissiveness seem less consistent [73]. Dual process frameworks have been proposed to understand both the reflective pathways (volitionally shaping a child’s food environment) and the automatic pathways (nonconscious or nonintentional processes) in which parenting practices work to influence children’s food intake [74,75]). Furthermore, it should be taken into account that both parents and children have active roles, which means that parenting practices are influenced by children’s behaviors.

The evidence regarding physical activity parenting practices is lagging behind that regarding food parenting practices [76]. However, systematic reviews have also provided evidence for the relation between parenting practices and physical activity and sedentary behaviors [59,62,76,77,78], although the effect of some parenting practices is less clear. For example, parental support (including logistic support and explicit role modeling) was found to be positively associated with children’s physical activity levels whereas the effects of restrictions and rules are less consistent [76]. Timperio and colleagues [79] found that the effect of some parenting practices (provision of equipment; financial, logistic, and emotional support; reinforcement) on children’s physical activity was in part mediated by children’s physical activity attitudes, beliefs, perceived behavioral control and enjoyment. However, conclusions regarding the causal chain are not possible. Furthermore, in light of the large spectrum of parenting practices this might depend on the specific parenting practice.

It should be noted that within this sector as well as within the other parent-child sectors (2b, 3b), both mother-child and father-child factors (e.g., mothers’ parenting practices, fathers’ parenting practices) can exert their own separate influence, but their co-occurrence has an impact as well [80]. However, co-parenting has rarely been addressed [81]. There is some evidence showing that mothers’ and fathers’ differ in their use of parenting practices and that maternal and paternal parenting behaviors (parenting practices, parenting styles) differ in their association with children’s EBRBs (e.g., [82,83,84]).

##### Family—Family Practices (1c)

We propose the term “family practices” to cover immediate family-level influences. In contrast to parenting practices, which are considered reflections of parent-child dyadic interactions, family practices reflect interaction patterns at the family level. Family practices could thus be regarded as behavior-specific manifestations of more general family-level factors that occur in daily family life. The best known example of family practices is that of family meal practices (e.g., frequency of family meals, eating with the television on). Berge [36] considered family meals to be a proxy for family interactions: the way families manage family meals is indicative of overall family functioning.


*Association with Children’s EBRBs*


There is convincing evidence that family meals are associated with a healthy dietary intake and a normal weight range [36,85]. Furthermore, family meals seem to serve as a protective factor against several risk factors among adolescents [86]. However, most studies have only examined the frequency of family meals, and not those of other components, so the underlying mechanisms of the protective effects of family meals are not yet understood [86]. Family meals are a context for family dynamics. Berge and colleagues [87] examined family-level food-related dynamics by using direct observational methods. Positive food-related dynamics during family meals (e.g., food communication, food warmth) were found to be associated with reduced prevalence of childhood obesity [87].

Family practices regarding physical activity or sedentary behavior have hardly been addressed. Some studies included family-based physical activities (e.g., [88]) or watching television as a family [63] along with other parental factors. Cleland and colleagues [88] found that family-based activities (e.g., co-participation in physical activities such as walking or cycling) were significantly associated with changes in moderate to vigorous physical activity (MVPA) over 5 years among girls. Regarding sedentary behavior, studies found that family co-viewing practices (watching TV as a family and playing games/computers as a family) were associated with children’s TV viewing [63,89].

#### 2.3.2. Proximal Level

##### Individual—Parents’ Behavior-Specific Characteristics (2a)

This sector of the LIFES framework comprises parents’ personal cognitive, motivational, volitional or affective factors relating to their own EBRBs (e.g., enjoyment of physical activity, self-efficacy for healthy eating, attitudes regarding EBRBs, intrinsic motivation). The model proposes that constructs at this level will not influence the children’s and adolescents’ EBRBs directly but via relevant mediators (e.g., children’s behavior specific characteristics or parents’ EBRBs or parental cognitions).


*Association with Children’s EBRB*


Parents’ behavior-specific factors are related to parental cognitions (which represent a variable of the parent-child subsystem, sector 2b). Naisseh and colleagues [90] showed that parental beliefs (perceived importance of their children’s physical activity and perceived competence of their children in physical activities) and parental support significantly differed across motivational profiles of parents (ranging from highly self-determined and moderately self-determined to non-self-determined, and externally motivated profiles). Parents in the highly self-determined cluster had high ratings of perceived importance and the highest ratings of perceived competence. These parental cognitions have been shown to be related to children’s physical activity (e.g., [91]).

Furthermore, parents’ enjoyment of physical activity has been shown to be related to their children’s motivation toward physical activity [92] and the extent to which their children engage in physical activity [93].

To the best of our knowledge, hardly any studies have focused on these factors in the contexts of nutrition or sedentary behavior.


*Example of Mediated Effects*


To the best of our knowledge there are not any studies examining mediated effects of parents’ behavior-specific characteristics (2a) on children’s EBRBs via children’s behavior specific characteristics or parents’ EBRBs (1a) or parental cognitions (2b).

##### Parent-Child—Behavior-Specific Characteristics of the Parent-Child Subsystem (2b)

In contrast to parents’ personal factors related to their own behavior, this sector comprises parental factors (factors reflecting the parent-child subsystem) relating to children’s behaviors. Examples include parental beliefs about children’s EBRBs, such as the perceived importance of the behavior for the children’s health. These factors are assumed to be antecedents of parenting practices. Parental beliefs are supposed to influence parent-child interaction patterns, including the extent of encouragement and the provision of opportunities and experiences that, in turn, affect their children’s EBRBs as well as children’s motivation [94].


*Associations with Children’s EBRB*


Parental beliefs are related to children’s exercise participation. Kimiecik and Horn [95] showed that parental beliefs regarding their child’s physical activity accounted for 27% of the variance in children’s MVPA, while mothers’ and fathers’ physical activity levels were not associated with children’s moderate to vigorous physical activity. Mutz and Albrecht [60] found that parents’ beliefs in sports’ capacity to foster personality development, social integration and character building were associated with children’s objectively measured MVPA. Using the Expectancy Value Model, Fredericks and Eccles [91] showed that parental competence and value beliefs are associated with children’s own beliefs (about competence and value) and their sports participation. Perceived competence is an important predictor of children’s physical activity, and parental beliefs are important for shaping children’s self-perceptions (e.g., [96]). Furthermore, parental beliefs and cognitions relate to parenting practices. For example, Trost and colleagues [97] found that the perceived importance of physical activity for their child was associated with parental support. In addition, parental cognitions regarding sedentary time have been shown to be related to children’s sedentary behavior. Andrews and colleagues [98] found that the more TV parents watched, the less likely they were to think that limiting TV viewing would keep their children healthy. However, according to the LIFES framework, this is assumed to be an indirect (mediated via parents’ individual-level attitudes regarding TV viewing, sector 2a) rather than a direct influence. In the nutrition context, studies showed that parents’ weight concerns regarding their child relate to parenting practices (e.g., [99,100]). Parental cognitions regarding the provision of healthy foods (attitudes, subjective norms, behavioral control and outcome expectancies) predicted the intention to monitor their children’s food intake and the degree to which parents’ actually monitored this [98]. Similar results obtained by Gevers and colleagues [101] and Baranowski and colleagues [102] underline the importance of parental cognitions for the use of different parenting practices.


*Example of Mediated Effects*


To the best of our knowledge, the indirect effect of behavior-specific characteristics of the parent-child subsystem such as parental beliefs (2b) on children’s EBRBs via parenting practices (1b) or children’s behavior specific factors (e.g., attitudes, intrinsic motivation) has rarely been studied. One example is, that Bois and colleagues [103] found that mothers’ beliefs about their child’s competence had an indirect effect on children’s physical activity by influencing children’s perceived competence which, in turn, contributed to children’s level of physical activity involvement. Interestingly, the indirect effect was not found for fathers’ beliefs [103].

##### Family—Behavior-Specific Characteristics of the Family as a Whole (2c)

This sector of the LIFES framework includes the equivalent of parental behavior-specific factors but at the family level, e.g., family-level behavior-specific cognitions, motivation etc. The model proposes that constructs at this level are indirectly related to children’s and adolescents’ EBRBs via relevant mediators (e.g., family practices or child’s motivation). To the best of our knowledge there is, so far, only one example of this idea in the literature: the Family Health Climate concept. The Family Health Climate (FHC) is defined as the shared perceptions and cognitions concerning a healthy lifestyle within a family [104]. It reflects the individual experience of daily family life, the evaluation of health-related topics and expectations with respect to typical values, behavior routines and interaction patterns within the family. The FHC serves as a framework for an individual’s everyday health behavior, forms the basis of regulating health-related behaviors and provides references for valuing and interpreting individuals’ own behavior and that of others. The FHC can be assessed regarding healthy eating using the FHC-Nutrition scale and regarding physical activity using the FHC-Physical Activity scale.


*Association with Children’s EBRB and Example of Mediated Effects*


The nutrition climate was found to correlate positively with general parenting and healthy food parenting practices [105]. Furthermore, it has been shown that the physical activity climate is related to the support provided and received within families, and to joint physical activities and that the perceived nutrition climate is associated with family meals and availability of vegetables and soft drinks [104]. In addition, the physical activity climate was found to affect physical activity behavior, while the nutrition climate affected the dietary behavior of adolescents [106]. These effects were mediated by adolescents’ intrinsic motivation.

#### 2.3.3. Distal Level

##### Individual—Parents’ General Factors (3a)

This sector comprises individual-level factors that reflect general characteristics of the parents. Examples are demographic variables such as educational level as well as psychological variables such as personality factors, need for cognition and IQ [107]. These factors are assumed to be indirectly related to children’s EBRBs or operate as higher-level moderators.


*Association with Children’s EBRB*


Studies showed that parents’ personality relates to parenting (sector 3b). A meta-analytic review investigating the association between the Big Five personality factors and parenting showed meaningful relations of higher levels of extraversion, agreeableness, conscientiousness and openness on the one hand and lower levels of neuroticism on the other with warmth and behavioral control, and of higher levels of agreeableness and lower levels of neuroticism with autonomy support [108]. Several studies related socio-economic status to children’s and adolescents EBRBs and found that a low socio-economic status was associated with poorer diets, less physical activity [109] and more sedentary behavior (e.g., [110]). LIFES assumes that this influence is not direct but indirect, via relevant mediators.


*Example of Mediated or Moderated Effects*


In line with this assumption, Vereecken and colleagues [111] found that parenting practices differed according to parents’ educational levels (3a) (mothers with higher educational level being less permissive, praising their children more often for the consumption of fruit and vegetables and abstaining from sweets themselves in the presence of their children) and that parenting practices mediated the relationship between parents’ educational level and children’s dietary intake. However, according to LIFES, the relationship between educational level and parenting practices is indirect, as well. Potential mediators could be knowledge regarding the impact of behavior on health, or concerns about health risks. (2a).

##### Parent-Child—General Parent-Child Characteristics (3b)

This sector comprises general parent-child interaction patterns which are not specific for EBRBs. It refers to general parent-child interactions such as the so-called parenting style [112] or general parenting. General parenting is the emotional context provided by parents, the broader context in which parent-child interactions take place. General parenting is often expressed by the extent to which parents are responsive, warm or nurturing to the child (responsiveness) and the extent to which parents control their children (restrictiveness). Parents who are both responsive and strict are called authoritarian [112]. Recently, a third dimension of parenting was recognized, next to warmth vs. rejection and autonomy support vs. coercion, namely structure vs. chaos [113,114], which concerns the extent to which parents organize their children’s environment and the children’s daily routines, and whether they react to the children in a consistent manner. Within these three dimensions, five key parenting constructs have been identified: Nurturance, Structure, Behavioral Control, Coercive Control and Overprotection (see [114] for more information). LIFES assumes that general parenting factors (3b) influence children’s EBRBs via a mediated route through their impact on behavior-specific parental cognition (2b) and subsequent parenting practices (1b), but also as a higher level moderator of the relationship between parenting practice and children’s EBRBs [30].


*Association with Children’s EBRB*


The body of evidence regarding general parenting and EBRBs is growing. Sleddens and colleagues [30] performed a systematic literature review on the evidence regarding nutrition, physical activity and parenting style. They concluded that, overall, parents who raise their children using an authoritative parenting style have healthier children, in terms of weight-related outcomes, compared to children raised in different styles. Additionally, Pinquart [115] conducted a meta-analysis and found that positive parent-child relationships and higher levels of parental responsibility were associated with healthier eating and more physical activity of their children.

The findings described above are likely to be explained by interaction effects between general parenting and parenting practices. As can also be seen in the LIFES framework, general parenting is a distal variable, relatively far removed from children’s health behaviors in the causal chain.


*Example of Mediated or Moderated Effects*


Some indications of moderating effects of general parenting have been found in different studies (e.g., [29,116,117]). For example, Sleddens and colleagues [117] found that for children who were reared in a positive parenting context (3b), the parenting practices of encouragement and covert control (1b) (keeping foods out of reach of children) were found to work better than for children who were raised in a different parenting context. A recent study of Lopez and colleagues [118] examined whether food-related parenting practices (1b) mediate the effect of parenting styles (3b) on children’s diet. They found that mealtime structure mediates the effect of authoritative, authoritarian and permissive parenting style on children’s healthy eating.

##### Family—General Characteristics of the Family as a Whole (3c)

Sector 3c comprises factors that characterize the family as a whole. Examples are general family functioning and family climate. Family functioning is a complex phenomenon describing the structure, organization and interaction patterns of the family unit [119]. A family’s structure and organization as well as interaction patterns shape individual behaviors of family members. Family functioning describes how families manage their daily routines and solve problems, how they communicate, how individuals fulfill their roles within the family, how the family responds to feelings, the degree to which the family is interested in and values activities of family members, and the level of behavioral control [119]. Family climate, which is a rather similar construct, is conceptualized according to Moos and Moos [120] along the dimensions of interpersonal relationships (e.g., cohesion, conflict), personal growth (e.g., independence, achievement orientations) and system maintenance (e.g., organization, control). According to LIFES these factors represent the overarching family context. Direct links to children’s and adolescents’ EBRBs are likely to be weak. Nevertheless, these factors are assumed to be important upstream factors that continuously exert their influence via indirect, mediated paths and as higher level moderators.


*Association with Children’s EBRB*


Studies have shown that family functioning is associated with children’s and adolescents’ EBRBs and weight status. A higher level of family functioning was associated with less sedentary behavior, higher intake of fruits and vegetables and breakfast consumption, and with more physical activity (only for boys) [39]. Similarly, Haines and colleagues [40] found associations with unhealthful behaviors (fast food intake, sugar-sweetened beverage intake, screen time, lack of physical activity and short sleep duration). Furthermore, higher family functioning decreases the likelihood that adolescents are overweight or obese, but only for girls [39,40]. The link with physical activity has been found for self-reported as well as objectively measured physical activity [121]. Family meals are supposed to be an indicator of family functioning [36]. Berge and colleagues [39] and Welsh and colleagues [122] found a significant correlation between frequency of family meals and family functioning (the latter used family cohesion as one dimension of family functioning). In line with this, family cohesion is associated with eating breakfast and soft-drink consumption [123].

Although the links with children’s and adolescents’ EBRBs were weak in the studies we examined, it is important to take the broader family context into account (e.g., [39,40,124]). These factors reflect the overarching family context in which these behaviors mostly take place, and are expected to indirectly influence children’s and adolescents’ EBRBs in various ways.


*Example of mediated or moderated effects*


To the best of our knowledge, these effects, e.g., family functioning (3c) as a moderator of the impact of family meals (1c) on dietary intake, has not been tested yet.

### 2.4. Translation of LIFES’ Assumptions into Potential Mediated and Moderated Pathways

Table 1 describes four possible pathways that could be derived from the LIFES framework. The assumed links are illustrated with possible factors from different sectors of the LIFES framework (see Figure 1). The first three described links hypothesize that the effect of level 2 factors (proximal level) on child’s EBRBs are mediated via level 1 factors (immediate level) or via equivalent level 2 child factors. For example, the 2c factor FHC (e.g., nutrition climate) is assumed to influence 1c family practices (e.g., frequency of family meals) and these family practices influence children’s dietary patterns. Furthermore, the effect of the FHC could be mediated via child factors on the proximal level, e.g., by influencing intrinsic motivation to engage in a specific EBRB. The last two pathways illustrate the higher-level moderation of factors on a higher level (e.g., level 3 factors moderating the impact of level 1 factors on child outcomes). LIFES hypothesizes for example that the strength of the association between parenting practices and child’s behavior depends on factors of the overarching family subsystem, e.g., family functioning.

## 3. Discussion

The importance of the family environment for children’s and adolescents’ EBRBs has been addressed in a large number of studies. Different types of influences of the family environment have been examined. Based on these studies and on theoretical approaches such as Family Systems Theories, the purpose of this paper was to describe the development of an overarching framework. The aims were (a) to bring together the different types and levels of influences in one framework, (b) to relate them to each other and (c) to postulate potential mechanisms and paths of influence. The core message is that family environmental influences arise from interactions between the individuals within the family, the parent-child subsystem and the family as whole. Furthermore, the influences can be differentiating according to their proximity to EBRBs (immediate, proximal and distal level). The LIFES framework covers the notion of the “family as a system” in that individuals’ behavior cannot be understood in isolation from the rest of the system. It takes into account children’s personal factors, parents’ individual factors, parent-child factors and factors at the overarching family level. The framework seeks to simplify the complexity of family influences on children’s EBRBs and illustrates which influences could (and should) be considered, where those influences are located, and how they are interrelated with other influences arising from the family. Potential direct and mediated links as well as moderators could be derived from this framework and be transmitted into distinct research models and testable hypotheses.

Researchers could focus on the interrelationship between factors within one sector (e.g., interaction between mother-child and father-child factors such as parenting styles, parental beliefs, parenting practices) and the relationships to children’s outcome variables. Furthermore, they could address how those factors are related to children’s EBRBs (e.g., mediators such as children’s motivation) and under what circumstances they are related (e.g., moderators such as family structure, family functioning).

While the importance of different family environmental influences for children’s and adolescents’ EBRBs has been shown in a large number of studies, overall there is a lack of studies focusing on the interplay between these different influences. Examining this interplay and considering mediation and moderation effects is crucial for a better understanding of the development and maintenance of healthy or unhealthy life styles.

For example, the model illustrates that small associations between distal variables (e.g., family functioning) and children’s EBRBs (e.g., [30,40]) should not be interpreted as irrelevant but as potentially important upstream factors that influence children’s behavior via mediated paths and as higher level moderators (e.g., [109,115]). For example, Kimiecik and Horn [125] concluded in their study, that parenting styles may reflect parent-child interaction patterns that impact the development of children’s physical activity beliefs, such as value and competence. A better understanding of the influences of different factors on children’s and adolescents’ EBRB and the integration of distal variables of different subsystems might be crucial for planning interventions, for example regarding the question how to involve parents and families in intervention programs [126,127]. Therefore, incorporating LIFES in the development of intervention programs could improve their effect on children’s EBRBs.

Recent studies have recommended considering the overarching family context [39,40,87,106]. It is especially the immediate and proximal levels within the family subsystem which have rarely been addressed. Even though several studies have addressed family meals, studies examining other family practices in the context of eating behavior, as well as studies addressing the context of physical activity and sedentary behavior, are lacking. A qualitative study revealed that “family activities” are deemed to be very important by families. Parents stated that these activities relate to family functioning and to benefits for children, e.g., development of health lifestyles [128]. As far as we know, the Family Health Climate is to date the only empirically investigated factor at the proximal level, in relation to children’s and adolescents’ EBRBs [104,106]. We suggest investigating variables that reflect the family subsystem at the immediate and proximal levels and especially regarding family practices related to physical activity (e.g., joint physical activities) and sedentary behavior (e.g., watching TV as a family).

Examining the interplay of different family environmental influences and taking into account influences from different levels and subsystems requires a consistent use of terminology. As Vaughn and colleagues [25] stated, the research field would benefit from solving inconsistencies in terminology and definitions. This becomes evident when factors from the parent-child subsystem and the family subsystem are addressed simultaneously. Factors from these two subsystems have to be clearly distinguished. An example are “family meals”, which are often described in terms of parenting practices even though they are actually a factor of the family subsystem (e.g., [36]). Therefore, it might help the field moving forward if researchers strive to find a consensus in terminology and definitions for example via establishing an expert group, implementing expert symposia and publishing the results (for example, see [129]).

Strictly speaking, the parent-child subsystem refers to the mother-child and father-child subsystems. It should be noted that the evidence regarding the important role parents play refers mainly to the mother-child subsystem [130]. Although fathers undoubtedly play an important role, it remains poorly understood what exactly this role looks like, if there are behavior-specific differences (in terms of physical activity, diet, sedentary behavior) or gender-specific differences (daughters vs. sons), how the father’s role differs from mother’s role, how mothers’ and fathers’ interact (e.g., co-parenting) and how this interaction affects child’s EBRBs [131]. It should be noted, that the term “parent” in the framework is not limited to natural parents. It could be applied to caregiving members within the family system including step-parents but also grandparents if applicable. Furthermore, family is a heterogeneous social context. Therefore, underlying family characteristics, such as single- and double-parent families, blended families or multi-generational families should be taken into account. Furthermore, the social and physical context the family is living in might influence family dynamics as higher-level moderators. However, to date, both empirical data and theoretical approaches are lacking to incorporate these issues in LIFES at this point. We encourage research regarding these questions.

LIFES seeks to cover different behaviors, in particular regarding physical activity, eating and sedentary behavior. The framework is the same, but the importance of specific factors and their interrelationships are supposed to differ depending on the behavior. The same applies to the age and gender of the children and adolescents. The relevance of specific factors might differ for children and adolescents and for boys and girls. For example, Haines and colleagues [40] showed gender differences in the association between family functioning and weight status. While a higher level of family functioning was associated with decreased likelihood of being overweight among girls, this was not the case for boys.

LIFES zooms in on the family context. This does not mean, however, that we would disagree with scholars advocating a more holistic view of environmental influences on EBRBs, for example by including the school environment [132] or by focusing on interactions between multiple microsystems in the mesosystem [133]. Indeed, taking into account other contexts is important; nevertheless, we have to gain knowledge how the specific setting or system, in this case the family, works and how interventions could be effectively implemented within this system. This could serve as a valuable precondition for realizing more effective upstream ecological approaches. However, in the end, the research question will determine which focus is most suitable.

## 4. Conclusions

LIFES is a theoretical framework that aims to organize different family environmental influences on children’s EBRBs. The framework highlights that family environmental influences arise from the individuals within the family, from the parent-child subsystem and from the family as whole. LIFES combines these three subsystems (individual, parent-child, family) with three levels of environmental factors ranging from behavior-specific to general (immediate, proximal, distal). It emphasizes reciprocal influences or feedback loops between the factors and includes both direct and mediated influences as well as higher-level moderation effects. It should be noted that the LIFES framework is not designed to be a model that should be tested as a whole. LIFES provides a framework to classify family environmental factors, to match different factors to adequate outcome variables and to examine the assumed direct, mediated and moderated effects arising from the family environment. It may support researchers to identify and include relevant variables in their research design and to study underlying mechanisms. Furthermore, targeting not only the child and the parent-child dyad but also taking into account the family as a whole and addressing factors on different levels will probably enhance the effectiveness of interventions. Finally, evaluation designs that are inspired by LIFES would aim to measure relevant processes and outcome indicators at different levels and in different subsystems.

## Figures and Tables

**Figure 1 ijerph-15-02714-f001:**
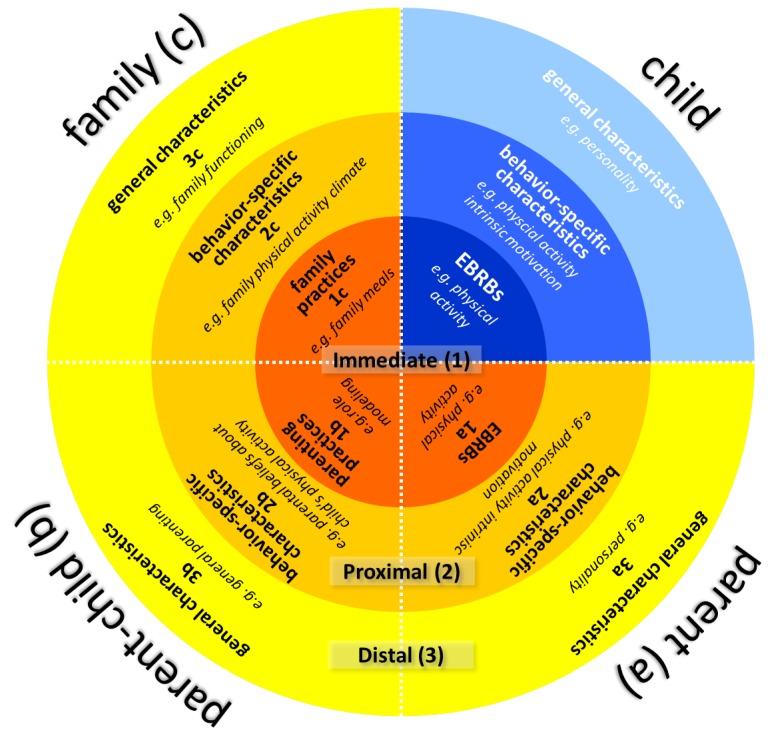
Levels of Interacting Family Environmental Subsystems (LIFES).

**Table 1 ijerph-15-02714-t001:** Possible factors and assumed pathways within the LIFES framework.

Pathways Derived from LIFES	Potential Factors and Assumed Pathways
2c-1c-1child	positive nutrition climate (2c) → more frequent family meals (1c) → child eats more regularly and has a higher intake of healthy foods (e.g., vegetables) (1child)
2c-2child-1child	positive physical activity climate (2c) → child has more intrinsic motivation to be physically active (2child) → child is more physically active (1child)
2a-2b-1b-1child	negative attitude regarding sugar sweetened beverages (2a) → more perceived importance that child does not drink sugar sweetened beverages (2b) → role modeling of drinking water instead of sugar sweetened beverages (1b) → child drinks more water instead of sugar sweetened beverages (1child)
3c moderates 1b-1child	higher cohesion within the family (3c) stronger impact (moderation) of role modeling of reducing sedentary time (1b) → child reduces sedentary time (1child)
3a-3b moderates 2b-2child-1child	higher level of agreeableness of a parent (3a) → more autonomy supportive parenting (3b) stronger impact (moderation) of more positive attitude towards child’s playing outside (2b) → more intrinsic motivation for playing outside in child (2child) → child plays more outside (1child)

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
