# Peer review of "Conceptualizing Family Influences on Children’s Energy Balance-Related Behaviors: Levels of Interacting Family Environmental Subsystems (The LIFES Framework)"

_ijerph, 2018, doi:10.3390/ijerph15122714_

Round 1
Reviewer 1 Report
The paper is interesting and has a careful and consistent layout. The Authors have developed the model how family environment affects children’s and adolescents’ energy balance-related behaviours. Their LIFES model includes three levels and three subsystems based on strong theoretical background. Additional assumption was made to consider interaction between individual elements. Each of the nine elements of the model has been described in detail. Numerous examples of papers matching these nine elements were provided. Attempts were made to give some examples of empirical research relating to nutrition and physical activity / sedentary behaviours. In Part 2b, papers on nutrition are placed inside the part on physical activity, which is the only example that disturbs the consistency of the whole design.The main doubt concerns the mentioned interaction thread. It seems to me that it has not been sufficiently emphasized. This is probably due to the fact that the effect of moderation or mediation is still rarely described. The article would be more consistent if, after each part of the Association with children's EBRB, some examples of interaction are mentioned. As alternative it should be noted that no paper with interaction was found.
It is also not clear why rather theoretical part 2.4. is at the end. It is not stated whether and how it results from the former literature review. This part could be presented as a continuation of the model description. The word "examples" in the title (2.4) suggests some references to published research presenting mechanisms and paths.
It also seems to me that at the end of conclusions there is too much discussion with the reference to literature. I suggest more concise conclusions.
Author Response
Point 1: In Part 2b, papers on nutrition are placed inside the part on physical activity, which is the only example that disturbs the consistency of the whole design.
Response 1: Thanks for the overall compliments and for this specific remark. We now moved the nutrition section to the end of the paragraph
Point 2: The main doubt concerns the mentioned interaction thread. It seems to me that it has not been sufficiently emphasized. This is probably due to the fact that the effect of moderation or mediation is still rarely described. The article would be more consistent if, after each part of the Association with children's EBRB, some examples of interaction are mentioned. As alternative it should be noted that no paper with interaction was found.
Response 2: Thank you very much for this remark and the suggestion to include a separate section with examples for the interaction. We included this section for all sectors, restructured the presented studies and added some information where necessary.
Point 3: It is also not clear why rather theoretical part 2.4. is at the end. It is not stated whether and how it results from the former literature review. This part could be presented as a continuation of the model description. The word "examples" in the title (2.4) suggests some references to published research presenting mechanisms and paths.
Response 3: We decided not to move this section to the general description of the model (2.2) as the factors/variables that are described in the table are explained in 2.3. Therefore, it would be difficult for the reader to understand the proposed links without having any information about the sectors and the variables that are located in the sectors. We changed the subtitle and hope that it is more clear now.
Point 4: It also seems to me that at the end of conclusions there is too much discussion with the reference to literature. I suggest more concise conclusions.
Response 4: Thank you very much for this suggestion. We shortened the Conclusions and reorganized the Discussion.
Reviewer 2 Report
This paper provides a nice synthesis of the effects of family influences on healthy lifestyles in children/teens, and puts forth a very interesting framework for future work in this area. I have the following comments for consideration by the authors.
Background:
In the first 2 paragraphs, the authors do a nice job of explaining what they aim to do in this paper. I think it would be helpful to have a comment about how this might help the field advance? How might this knowledge better inform future interventions? I think this would help the reader be more motivated to continue reading.The authors group together children and adolescents in their explanations. However, I believe there could be very different forces at play through the child’s life course, things that would be important to take into account when looking at the family context (such as adolescents having the ability to procure some of their own food, perhaps having a school or social schedule that precludes eating with the family, etc.). I believe it’s important at least for the authors to explain that these age groups are not the same, and justify why for this paper they feel it is appropriate to explore the family context keep this age group as one large group.
I’m curious about the narrow definition of family, exploring mainly parent-child interactions. How about other adults in the household (such as a live-in nanny) that might not be part of the traditional “family” but function as such in many settings? Also, what if the child has two active families within s/he is a part?
I acknowledge that this framework does not deal with the sibling subsystem. However, are there aspects to the family-child context that are affected by having multiple siblings, siblings with special needs, etc. that should be taken into account? I understand that decisions have to be made about where to draw the line, but I would appreciate at least a mention of this complexity.
I appreciate the inclusion of family sociodemographic factors. I wonder about the context in which the family is surrounded – for example, how their SES compares with that of the majority of other families attending the same school or in the same neighborhood. Outside of those as structural determinants, I would appreciate hearing the authors’ thoughts on whether those would affect an individual family’s dynamics.
Discussion/Conclusions:The authors mention here the variability in family structures, but they stop short of explaining why this wasn’t made more explicit in their model, and how the model might need to be modified with certain family structures. I would appreciate hearing more about this.
The authors state that there are other environments (school, etc.) that could be important as well. I would appreciate a more up-front discussion about why researchers should then focus on LIFES itself. How does a framework that hones in on just the family context help the field, and how can interventions be designed using this framework if these broader contexts are not included? I do believe there are reasons that this should be done, and I would appreciate the authors making this more explicit.
Author Response
Point 1: In the first 2 paragraphs, the authors do a nice job of explaining what they aim to do in this paper. I think it would be helpful to have a comment about how this might help the field advance? How might this knowledge better inform future interventions? I think this would help the reader be more motivated to continue reading.
Response 1: Thank you very much for the overall compliments and for this useful remark. We added the following paragraph: “LIFES aims to organize the variety of different family environmental factors and environment – behavior pathways in youth. It helps to advance the research field as it supports researchers to derive specific hypotheses regarding underlying mechanisms of family environmental influences on individual’s behaviors. LIFES-based research might inform future interventions as it reveals processes that impede or support individual behavior change. It could also help to find answers for example regarding the question how the family environment should be integrated in intervention programs in order to increase sustained effects on children’s and adolescents’ EBRBs.” (lines 56-63)
Point 2: The authors group together children and adolescents in their explanations. However, I believe there could be very different forces at play through the child’s life course, things that would be important to take into account when looking at the family context (such as adolescents having the ability to procure some of their own food, perhaps having a school or social schedule that precludes eating with the family, etc.). I believe it’s important at least for the authors to explain that these age groups are not the same, and justify why for this paper they feel it is appropriate to explore the family context keep this age group as one large group.
Response 2: Thank you; the life-course perspective indeed remains to be important. We now added “It should be noted that although the family environment is important for both children and adolescents, the relevance of specific family environmental factors will depend on the age of the child. Therefore, these age groups should be studied separately, or, at least, age should be incorporated not only as a confounder but also as a potential moderator.” (lines 94-98)
Point 3: I’m curious about the narrow definition of family, exploring mainly parent-child interactions. How about other adults in the household (such as a live-in nanny) that might not be part of the traditional “family” but function as such in many settings? Also, what if the child has two active families within s/he is a part? I acknowledge that this framework does not deal with the sibling subsystem. However, are there aspects to the family-child context that are affected by having multiple siblings, siblings with special needs, etc. that should be taken into account? I understand that decisions have to be made about where to draw the line, but I would appreciate at least a mention of this complexity.
Response 3: Thank you very much. We added: “It should be noted that family is an inhomogeneous social context, there is no narrow definition in this research field. A family could be described as a complex interacting system and an organized whole. This description includes natural parents as well as other caregiving members such as step-parents or grandparents as part of the family system. Families also incorporate siblings and there is growing evidence regarding the important role of sibling relationships (e.g. [51–53]).” (lines 169-173)
Point 4: I appreciate the inclusion of family sociodemographic factors. I wonder about the context in which the family is surrounded – for example, how their SES compares with that of the majority of other families attending the same school or in the same neighborhood. Outside of those as structural determinants, I would appreciate hearing the authors’ thoughts on whether those would affect an individual family’s dynamics.
Response 4: Please see the answer to point 5
Point 5: The authors mention here the variability in family structures, but they stop short of explaining why this wasn’t made more explicit in their model, and how the model might need to be modified with certain family structures. I would appreciate hearing more about this.
Response 4+5: Thank you for this very interesting question. As empirical data as well as theoretical work in this direction is lacking we cannot make assumptions regarding this issue. We added: “Furthermore, it should be noted that family is a heterogeneous social context. Therefore, underlying family characteristics, such as single- and double parent families, blended families or multi-generational families should be taken into account. Furthermore, the social and physical context the family is living in might influence family dynamics as higher-level moderators. However, to date, both empirical data and theoretical approaches are lacking to incorporate these issues in LIFES at this point. We encourage research regarding these questions.” (lines 565-570)
Point 6: I agree with the authors that consistency across studies in constructs and terminology are needed. Are there specific recommendations that the authors have here that might help guide the field moving forward?
Response 6: We added: “Therefore, it might help the field moving forward if researchers strive to find a consensus in terminology and definitions for example via establishing an expert group, implementing expert symposia and publishing the results (for example, see [130].” (lines 553-555)
Point 7: The authors state that there are other environments (school, etc.) that could be important as well. I would appreciate a more up-front discussion about why researchers should then focus on LIFES itself. How does a framework that hones in on just the family context help the field, and how can interventions be designed using this framework if these broader contexts are not included? I do believe there are reasons that this should be done, and I would appreciate the authors making this more explicit
Response 7: We added "Indeed, taking into account other contexts is important; nevertheless, we have to gain knowledge how the specific setting or system, in this case the family, works and how interventions could be effectively implemented within this system. This could serve as a valuable precondition for realizing more effective upstream ecological approaches." (lines 582-585)
Reviewer 3 Report
The authors have provided a useful compilation of the many factors that influence child health and other outcomes. The theoretical model seems relevant and the research review is adequate in supporting the model. It is possible that this model could serve to generate more comprehensive research on these issues. It is well written and conclusions do not go beyond the authors' theoretical ideas.
Author Response
Thank you very much for the overall compliments.
Round 2
Reviewer 2 Report
I thank the authors for their edits/revisions. I believe the manuscript is now much clearer and makes an interesting contribution to the literature.